# The Role of Donor Gamma-Glutamyl Transferase as a Risk Factor for Early Graft Function after Liver Transplantation

**DOI:** 10.3390/jcm12144744

**Published:** 2023-07-18

**Authors:** Quirino Lai, Fabio Melandro, Tommaso M. Manzia, Gabriele Spoletini, Anna Crovetto, Gaetano Gallo, Redan Hassan, Gianluca Mennini, Roberta Angelico, Alfonso W. Avolio, Frederik Berrevoet, Luís Abreu de Carvalho, Salvatore Agnes, Giuseppe Tisone, Massimo Rossi

**Affiliations:** 1General Surgery and Organ Transplantation Unit, Department of General and Specialty Surgery, Sapienza University of Rome, AOU Policlinico Umbertot I of Rome, 00185 Rome, Italy; fabio.melandro@uniroma1.it (F.M.); gaetano.gallo@uniroma1.it (G.G.); redan.hassan@uniroma1.it (R.H.); gianluca.mennini@uniroma1.it (G.M.); massimo.rossi@uniroma1.it (M.R.); 2HPB and Transplant Unit, Department of Surgical Sciences, University of Rome Tor Vergata, 00173 Rome, Italy; tomanzia@libero.it (T.M.M.); roberta.angelico@uniroma2.it (R.A.); tisone@uniroma2.it (G.T.); 3General Surgery and Liver Transplantation, Fondazione Policlinico Universitario Agostino Gemelli IRCCS, 00168 Rome, Italy; gabriele.spoletini@gmail.com (G.S.); alfonso.avolio@unicatt.it (A.W.A.); salvatore.agnes@policlinicogemelli.it (S.A.); 4Department of General and HPB Surgery and Liver Transplantation, Ghent University Hospital, 9000 Ghent, Belgium; anna.crovetto@me.com (A.C.); frederik.berrevoet@ugent.be (F.B.); luis.abreudecarvalho@uzgent.be (L.A.d.C.)

**Keywords:** expanded-criteria donor, graft loss, liver transplantation, donor risk index

## Abstract

Background: Growing interest has been recently reported in the potential detrimental role of donor gamma-glutamyl transferase (GGT) peak at the time of organ procurement regarding the risk of poor outcomes after liver transplantation (LT). However, the literature on this topic is scarce and controversial data exist on the mechanisms justifying such a correlation. This study aims to demonstrate the adverse effect of donor GGT in a large European LT cohort regarding 90-day post-transplant graft loss. Methods: This is a retrospective international study investigating 1335 adult patients receiving a first LT from January 2004 to September 2018 in four collaborative European centers. Results: Two different multivariable logistic regression models were constructed to evaluate the risk factors for 90-day post-transplant graft loss, introducing donor GGT as a continuous or dichotomous variable. In both models, donor GGT showed an independent role as a predictor of graft loss. In detail, the log-transformed continuous donor GGT value showed an odds ratio of 1.46 (95% CI = 1.03–2.07; *p* = 0.03). When the donor GGT peak value was dichotomized using a cut-off of 160 IU/L, the odds ratio was 1.90 (95% CI = 1.20–3.02; *p* = 0.006). When the graft-loss rates were investigated, significantly higher rates were reported in LT cases with donor GGT ≥160 IU/L. In detail, 90-day graft-loss rates were 23.2% vs. 13.9% in patients with high vs. low donor GGT, respectively (log-rank *p* = 0.004). Donor GGT was also added to scores conventionally used to predict outcomes (i.e., MELD, D-MELD, DRI, and BAR scores). In all cases, when the score was combined with the donor GGT, an improvement in the model accuracy was observed. Conclusions: Donor GGT could represent a valuable marker for evaluating graft quality at transplantation. Donor GGT should be implemented in scores aimed at predicting post-transplant clinical outcomes. The exact mechanisms correlating GGT and poor LT outcomes should be better clarified and need prospective studies focused on this topic.

## 1. Introduction

Liver transplantation (LT) represents the best curative therapy for a large number of acute and chronic end-stage liver diseases [1] and hepatic primary and secondary tumors [2,3]. Unfortunately, many patients waiting for an LT die during the waiting period due to the scarcity of available grafts [4]. Consequently, different strategies have been adopted worldwide with the intent to extend the donor pool. Among them, the use of living donation [5], aged donors [6], deceased cardiac donors [7], and non-standard donors due to infective [8] or tumoral [9] characteristics can be reported.

However, it has been documented that adopting expanded-criteria donors should be correlated with an increased risk of poor post-LT outcomes [10]. Therefore, an accurate selection of the parameters identifying high-risk donors for early graft loss after transplantation is of paramount relevance. Several scores have been developed based on recipient-related [11] or donor-related characteristics [10], or on a combination of both [12,13].

Recently, growing interest has been focused on the potential detrimental role of donor gamma-glutamyl transferase (GGT) peak at the time of organ procurement regarding the risks of graft discard [14] and poor outcomes after transplantation [15,16,17]. In the present study, we hypothesized that high donor GGT values had a relevant predictive value for increased 90-day post-transplant graft-loss risk. The study aims to demonstrate this adverse effect of donor GGT in a large European LT cohort.

## 2. Materials and Methods

### 2.1. Study Design

This is a retrospective international study based on prospective databases of patients receiving a first liver transplantation. The primary study endpoint was to investigate whether high donor GGT values were correlated with reduced post-LT graft survival.

The study followed the Strengthening the Reporting of Observational Studies in Epidemiology (STROBE) reporting guidelines [18]. All authors had access to the study data and reviewed and approved the final manuscript. The institutional review board of Azienda Ospedaliero-Universitaria Policlinico Umberto I (coordinating center; approval number 507/2021) approved the study.

### 2.2. Setting

Participants included four centers: Sapienza University Rome, Italy (*n* = 384); Tor Vergata University Rome, Italy (*n* = 364); Catholic University Rome, Italy (*n* = 339); and Ghent University Hospital, Belgium (*n* = 309). A total of 1335 patients were enrolled. All the centers involved did not present a specific common protocol in the acceptance of donors with high GGT values.

### 2.3. Population

All enrolled cases were adults (≥18 years), consecutively receiving a first LT from January 2004 to September 2018. Exclusion criteria were pediatric transplant patients (age < 18 years) and retransplantation. All the cases investigated included only LTs performed using grafts from deceased donors. No living donors or domino transplants were considered in the analysis.

### 2.4. Variables and Data Collection

Data collected in the study included recipient age; HCC, acute liver failure; HCV; HBV; alcohol-related cirrhosis; NASH, MELD; cold ischemia time; regional vs. extra-regional graft share; distance LT center procurement; use of airplane by the procurement team; donor age; donor sex; split/partial liver; anoxia as the cause of donor death; cerebrovascular accident as the cause of donor death; length of donor ICU stay; donor BMI; donor history of diabetes; donor history of arterial hypertension; previous donor surgery; donor history of tobacco abuse; donor history of alcoholic abuse; donor anticore positivity; donor hemodynamic instability before procurement; cardiac arrest in the donor before procurement; vasoactive score (VAS); donor sodium peak (mEq/L); donor AST peak (IU/L); donor ALT peak (IU/L); donor total bilirubin peak (mg/dL); donor INR peak; donor platelets peak; and last available donor GGT at the time of procurement (IU/L). All the donor blood tests collected during the ICU stay were evaluated, and the peak values were collected. In the case of GGT, the last available value was considered. The donor GGT peak value was not collected due to a high number of missing data.

### 2.5. Definitions

Ninety-day graft loss was defined as any episode of loss of liver function due to patient death, re-listing, or retransplantation reported within three months of liver transplantation. The last follow-up date was 31 March 2021.

### 2.6. Statistical Analysis

Continuous variables were reported as medians and the first–third quartile (Q1–Q3). Dummy variables were reported as numbers and percentages. For each variable, missing data involved less than 10%, being handled using the maximum-likelihood estimation method. The Mann–Whitney U test was used for comparisons between groups in the case of continuous variables, and Fisher’s exact text was adopted in the case of categorical variables.

The best cut-off of donor GGT for the risk of 90-day graft loss was established using ROC curves and the Youden index.

Multivariable logistic regression models were constructed using the backward Wald method to identify the risk factors for 90-day graft loss, high donor GGT, and macrovesicular steatosis at the time of procurement. Beta-coefficients, standard errors, odds ratio (OR), and 95% confidence intervals (95% CIs) were reported.

The additive predictive ability of donor GGT in terms of risk of 90-day graft loss was calculated using the Akaike information criterion (AIC). The AIC represents an estimator of prediction error and thereby the relative quality of statistical models for a given data set. In other terms, the lowest AIC value identifies the best model, namely the model that best fits reality. The AIC is calculated using the following formula: AIC = 2 K − 2 ln(L)
where K is the number of independent variables used and L is the log-likelihood estimate. If the delta AIC of a model is more than 2 AIC units lower than another, then it is considered significantly better than that other model. The AIC was calculated for commonly adopted scores of post-transplant graft survival prediction, such as the model for end-stage liver disease (MELD), donor age MELD (D-MELD), donor risk index (DRI), and balance of risk (BAR), as well as for these scores + donor GGT. Comparison among the different models was calculated, estimating the delta-AIC.

Kaplan–Meier survival curves were used to estimate graft and patient survival rates. Log-rank analysis was conducted to compare sub-groups. A *p*-value < 0.05 was considered statistically significant in all analyses. Statistical reports and plots were performed using the SPSS statistical package version 27.0 (SPSS Inc., Chicago, IL, USA).

## 3. Results

### 3.1. Characteristics of the Explored Cohort

A total of 1335 transplanted patients were enrolled. Adopting the Youden index, the best cut-off of donor GGT value at the time of procurement for the risk of 90-day graft loss was established using the ROC curves, corresponding to 160 IU/L. The cohort was split into two groups according to this value: low donor GGT (i.e., <160 IU/L; *n* = 1210, 90.6%) group or high donor GGT (i.e., ≥ 160 IU/L; *n* = 125, 9.4%) group. Donor-related characteristics in the investigated population are reported in Table 1. Recipient- and transplant-related characteristics are reported in Table 2.

As for the recipient- and transplant-related features, no significant differences were reported when comparing the two groups. In detail, no differences were reported in terms of recipient age, the main indication for transplantation, MELD laboratory score at the time of transplantation, and cold ischemia time. Conversely, some differences were reported when the donor-related characteristics were compared. In detail, the patients in the high donor GGT group had donors with a longer ICU stay (median 8 vs. 3 days, *p* < 0.0001), younger age (median 48 vs. 54 years; *p* = 0.02), and more donors with anoxia as the cause of death (12.0 vs. 6.8%, *p* = 0.04). Donor history of alcoholic abuse only merged statistical relevance, with more cases reported in the high donor GGT group (10.4 vs. 5.8%, *p* = 0.051). Donors with high GGT at procurement had more episodes of cardiac arrest before procurement (22.4 vs. 15.2%, *p* = 0.04) and a lower median VAS score (7 vs. 10, *p* = 0.01). As for blood analysis, median transaminase peaks were higher in the donors with higher GGT (*p* < 0.0001). Median total bilirubin peak merged significance (0.8 vs. 0.7 mg/dL, *p* = 0.055).

### 3.2. Risk Factors for 90-Day Graft Loss after Liver Transplantation

The role of different transplant- and donor-related features was investigated to explore the predictors for 90-day graft loss. Two different multivariable logistic regression models were constructed, introducing donor GGT as a continuous versus dichotomous variable (Table 3). In both models, donor GGT showed an independent role as a predictor of graft loss. In detail, the log-transformed donor GGT value was a risk factor for graft loss, with an OR = 1.46 (95% CI = 1.03–2.07; *p* = 0.03). The most relevant independent risk factors for graft loss were MELD (OR = 1.04, *p* < 0.0001) and donor total bilirubin peak (OR = 1.33, *p* < 0.0001), followed by the use of a split/partial liver, donor history of alcoholic abuse, donor age, and VAS score.

When the donor GGT value measured at the time of procurement was dichotomized according to the previously identified cut-off of 160 IU/L, its independent predictive role with an OR = 1.90 (95% CI = 1.20–3.02; *p* = 0.006) was confirmed. Also in this model, MELD was confirmed to be the most relevant risk factor for graft loss (OR = 1.04; *p* < 0.0001).

### 3.3. Additive Prognostic Role of Donor GTT

With the intent to explore the donor GGT role in predicting early graft loss after liver transplantation, this variable was added to the scores conventionally used for this intent (i.e., MELD, D-MELD, DRI, and BAR score). In all cases, when the conventional score was combined with donor GGT at the time of procurement, an improvement in the model accuracy was observed compared to the conventional model in which the score was evaluated by itself. In detail, adding donor GGT to the MELD score achieved the best model to fit with reality, with an AIC = 2179.50. All the other scores presented greater AIC values, therefore showing an inferior accuracy (the greater the AIC value, the lower the accuracy) (Table 4).

### 3.4. Parameters Correlated with High Donor GGT and Graft-Loss Rates According to Donor GGT Values

According to the data reported in Table 5, two independent donor-related variables correlated with the risk of elevated donor GGT (i.e., ≥160 IU/L), namely a long donor ICU length of stay (OR = 1.07, 95% CI = 1.05–1.10; *p* < 0.0001) and a previous history of donor alcohol abuse (OR = 1.92, 95% CI = 1.02–3.61; *p* = 0.04). Donor age only merged statistical significance as a risk factor for high donor GGT (OR = 0.99, 95% CI = 0.98–1.00; *p* = 0.053).

When graft-loss rates were investigated, significant superior rates were reported in LT cases with donor GGT ≥160 IU/L. In detail, 30-, 60-, and 90-day graft-loss rates were 17.6%, 20.8%, and 23.2% vs. 8.7%, 11.9%, and 13.9% in patients with high vs. low donor GGT, respectively (log-rank *p* = 0.004) (Figure 1).

### 3.5. Parameters Correlated with Macrovesicular Steatosis

A sub-analysis was performed in all donors with available biopsy at the time of procurement (*n* = 461), intending to investigate the risk factors for macrovesicular steatosis > 30%. According to the data reported in Table 5, three independent donor-related variables correlated with the risk of elevated macrosteatosis, namely donor BMI (OR = 1.15, 95% CI = 1.04–1.28; *p* = 0.006), duration of donor ICU length of stay (OR = 0.79, 95% CI = 0.67–0.94; *p* = 0.006), and donor GGT (OR = 1.004, 95% CI = 1.00–1.01; *p* = 0.044). Donor history of type 2 diabetes and bilirubin peak only showed a trend towards statistical significance.

## 4. Discussion

In the present study, donor GGT measured at the time of organ procurement was a relevant risk factor for post-LT graft loss, showing an additive prognostic role when added to the scores conventionally used to evaluate post-transplant outcomes. GGT represents a key transferase performing relevant roles in antioxidant defense mechanisms. Given its cellular role in antioxidant function, it has been investigated as a surrogate biomarker of oxidative stress [19]. Analogously, GGT may also have a pro-oxidant role, being very sensitive for diagnosing liver injury, although it has poor specificity for particular etiologies [20]. Consequently, a rise in GGT values might represent two antithetical mechanisms: liver regeneration and liver damage. Moreover, GGT has also been found to predict mortality across a spectrum of non-hepatic pathologies (i.e., metabolic and cardiovascular diseases, chronic kidney failure, and tumors) [21].

In the specific setting of liver transplantation, the role of donor GGT in predicting poor graft outcomes is controversial. Few studies have explored the role of donor GGT values in detail (Table 6) [14,15,16,17].

A study based on US data (*N* = 53,966) showed that donor GGT was significantly associated with an increased risk of liver discard, with GGT >200 IU/L showing a hazard ratio (HR) of 2.74 (95% CI = 2.51–2.99; *p* < 0.001). In the sub-group of transplanted cases (*N* = 47,249), GGT < 20 IU/L was a protective factor for the risk of graft loss (HR = 0.91, 95% CI = 0.83–1.00; *p* = 0.045) [14].

Of great interest is two recently published studies from North America and Europe explicitly exploring the risk of graft discard that did not incorporate this marker, pointing out the potential risk of underestimating its role in international databases [22,23].

A significant European analysis (*N* = 5723) intended to develop the Eurotransplant DRI, in which the classical parameters composing the American DRI [10] were integrated with a rescue allocation and the latest donor GGT value (HR = 1.06, 95% CI = 1.02–1.11; *p* = 0.005). After having integrated these variables, the predictive ability for the risk of graft loss significantly increased when the Eurotransplant vs. the American DRI were compared (c-indices: Eurotransplant-DRI 0.624 vs. DRI 0.614) [15].

A study from Germany (*N* = 678) investigated the impact of donor GGT in terms of post-LT early allograft dysfunction (EAD); by adding donor GGT (HR = 1.002; *p* = 0.047) to donor BMI, macrovesicular steatosis, and cold ischemia time, an equation was created to enable the prediction of EAD with a c-index of 0.68 [16].

An experience from France (*N* = 1152) explored the impact of donor GGT on the risk of 90-day graft failure, identifying a cut-off of 170 IU/L. In detail, the authors observed an increased risk of graft loss (39% at one year) in patients receiving livers from donors with a history of alcohol abuse and GGT ≥ 170 IU/L [17].

These experiences align with our results, in which a donor GGT ≥160 IU/L correlated with an increased risk of 90-day graft loss (OR = 1.90, 95% CI = 1.20–3.02; *p* = 0.006). As reported, adding the donor GGT to previous scores improved the concordance for diagnosing worse outcomes after transplantation.

The exact mechanisms justifying the correlation between donor GGT and poor post-LT outcomes are still under debate. Some studies have explored in detail the potential causes of this connection.

A recent Finnish study found that donor GGT predicted macrovesicular steatosis. Different GGT cut-offs were identified in donors without a history of alcohol abuse: a value of 66 IU/L had a sensitivity = 76% and a specificity = 68% for detecting macrosteatosis > 30%, while a value of 142 IU/L presented a sensitivity of 66% and a specificity of 83% for the detection of macrosteatosis > 60%. Among donors with alcohol abuse, a GGT value > 90 IU/L showed 100% sensitivity for detecting macrosteatosis > 60% [24].

These data are in line with our results, in which a correlation between macrosteatosis and increased GGT was reported: when investigated in biopsied grafts as an independent risk factor for a macrosteatosis > 30%, donor GGT appeared to be a relevant risk factor (OR = 1.004; *p* = 0.044).

A study from Canada on metabolomics and protein expression was performed in livers immediately after implantation. In this analysis, performed on 9 living donor LTs and 13 LT receiving a graft from a heart-beating donor, GGT was significantly increased in the immediate post-reperfusion phase of the livers coming from deceased donors. These livers suffered a longer cold ischemia time and performed worse than the living donor grafts during the neo-phase, demonstrating that GGT could serve as a sensitive index of early graft function [25].

A study from Switzerland based on 89 pediatric LTs receiving grafts from deceased donors showed a correlation between a longer donor ICU stay and increased GGT [26]. We also observed a similar result, with a median ICU length of 8 vs. 3 days (<0.0001) in LT recipients with donor high vs. low GGT at the time of procurement. In our series, ICU length of stay was also the most relevant risk factor for a donor GGT ≥ 160 IU/L (OR = 1.07, 95% CI = 1.05–1.10; *p* < 0.0001). It is not completely clear why such a correlation exists. Donors with a prolonged ICU stay are often excluded from organ donation because of a supposed deleterious effect of a lengthy ICU stay. We can only assume that different mechanisms might be involved, like an increased risk of infection, hemodynamic instability, and vasoconstriction derived from inotropic drugs, all concurring in a deleterious ischemic effect. Further studies are needed to clarify these aspects better.

The potential clinical implications of the findings reported in the present study should be of relevance in terms of pre-transplant evaluation and risk stratification. In fact, donor GGT is typically not used as a selection parameter for donor graft quality, and many scores used in the selection process avoid incorporating it. On the opposite, our study showed that the integration of donor GGT into commonly used scores plays a role in improving the risk stratification of the recipients. Therefore, a proposal for further exploring donor GGT in this setting should be of great relevance.

Moreover, the use of donor GGT as a relevant parameter to consider during donor evaluation should modify the ordinary management approach, requiring a graft biopsy per protocol in the presence of multiple risk factors (i.e., high GGT plus history of alcohol abuse). In our series, approximately 10% of our donors showed high GGT, with 54/125 (43.2%) of grafts selected after a biopsy. In all cases of a history of alcohol abuse, the biopsy was used for selecting the graft parenchyma quality. Further analyses on larger populations are required to investigate in detail the role of biopsy in the selection process of donors with high GGT.

The study presents some limitations. First, the study is retrospective and multicentric, presenting some selection-bias risk. Second, no protocol biopsies were performed in all the grafts at the time of procurement, limiting our analysis of the correlation between GGT and macrosteatosis. However, the biopsies were performed in the real-life clinical routine only when some doubt about graft quality was raised. No data were available on the initial donor GGT value, limiting the possibility of investigating a potential effect of GGT rising during the donor ICU stay. No analyses on metabolomics or proteomics were available on the donor and liver immediately after implantation. Unfortunately, no research in the literature exists on this specific topic. Further prospective studies are needed to better clarify the mechanisms connected between a GGT rise and liver damage. Lastly, a possible bias for generalizability exists. In fact, the increased rate of donors with a high GGT value may vary in other countries in accordance with the varying rates of alcohol use, NAFLD, and high BMI. Therefore, further studies coming from other countries are needed to confirm the impact of the donor GGT in other geographical settings.

In conclusion, donor GGT could represent a valuable marker for evaluating graft quality during transplantation. GGT should be implemented in the scores for predicting post-transplant clinical courses. GGT was correlated with more severe macrosteatosis and typically rose in donors with extended ICU stay. The exact mechanisms correlating GGT and poor LT outcomes should be better clarified, and there is a need for prospective studies focusing on this topic. Based on the current results, further exploration of the underlying mechanisms and clinical implications would contribute to a better understanding of the role of donor GGT in liver transplantation, potentially leading to improved outcomes for transplant recipients.

## Figures and Tables

**Figure 1 jcm-12-04744-f001:**
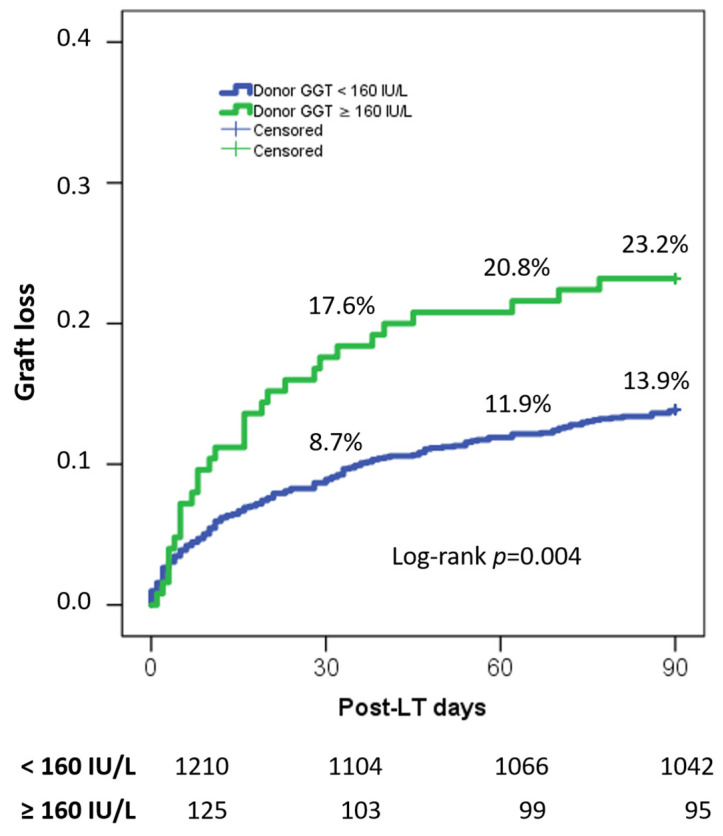
Graft-loss rates reported in patients transplanted receiving grafts from donors with high (≥160 IU/L) vs. low (<160 IU/L) GGT values.

**Table 1 jcm-12-04744-t001:** Donor-related characteristics in the investigated population.

Variables	Donor GGT < 160 IU/L (*n* = 1210, 90.6%)	Donor GGT ≥ 160 IU/L (*n* = 125, 9.4%)	*p*
	Median (Q1–Q3) or *n* (Percentage)		
Donor age, years	54 (38–66)	48 (33–61)	0.02
Donor male sex	655 (54.1)	68 (54.4)	1.00
Split liver	38 (3.1)	3 (2.4)	1.00
Trauma as cause of death	323 (26.7)	36 (28.8)	0.60
Anoxia as cause of death	82 (6.8)	15 (12.0)	0.04
CVA as cause of death	766 (63.3)	71 (56.8)	0.34
ICU length of stay, days	3 (2–5)	8 (5–10)	<0.0001
BMI	25 (23–28)	25 (24–28)	0.17
DM2	88 (7.3)	9 (7.2)	1.00
Arterial hypertension	413 (34.1)	37 (29.6)	0.32
Previous surgery	471 (38.9)	38 (30.4)	0.07
Smoking	396 (32.7)	50 (40.0)	0.11
Alcoholic abuse	70 (5.8)	13 (10.4)	0.051
Anticore positivity	108 (8.9)	11 (8.8)	1.00
Hemodynamic instability	387 (32.0)	41 (32.8)	0.84
Cardiac arrest	184 (15.2)	28 (22.4)	0.04
VAS	10 (3–24)	7 (0–20)	0.01
GGT measured at procurement, IU/L	27 (15–59)	256 (199–339)	<0.0001
Peak value			
Sodium, mEq/L	150 (145–156)	152 (146–157)	0.25
ALT, IU/L	39 (26–75)	74 (45–133)	<0.0001
AST, IU/L	31 (19–55)	82 (45–132)	<0.0001
Total bilirubin, mg/dL	0.7 (0.5–1.0)	0.8 (0.5–1.4)	0.055
INR	1.20 (1.08–1.31)	1.15 (1.06–1.25)	0.01
Platelets, ×10^3^/mcL	160 (118–218)	199 (153–306)	<0.0001

Abbreviations: GGT, gamma-glutamyl transferase; *n*, number; Q1–Q3, first–third quartile; CVA, cerebrovascular accident; ICU, intensive care unit; DM2, diabetes mellitus type 2; VAS, vasoactive score; ALT, Alanine transaminase; AST, aspartate transaminase; INR, international normalized ratio.

**Table 2 jcm-12-04744-t002:** Recipient- and transplant-related characteristics in the investigated population.

Variables	Donor GGT < 160 IU/L (*n* = 1210, 90.6%)	Donor GGT ≥ 160 IU/L (*n* = 125, 9.4%)	*p*
	Median (Q1–Q3) or *n* (Percentage)		
Recipient age, years	56 (49–62)	56 (47–62)	0.30
HCC	453 (37.4)	44 (35.2)	0.70
Acute liver failure	70 (5.8)	6 (4.8)	0.84
Cirrhosis			
HCV-related	366 (30.2)	43 (34.4)	0.36
HBV-related	153 (12.6)	19 (15.2)	0.4
Alcohol-related	420 (34.7)	46 (36.8)	0.69
NASH-related	98 (8.1)	10 (8.0)	1
MELD (laboratory)	16 (11–22)	17 (12–24)	0.33
CIT, min	410 (355–480)	420 (361–501)	0.20
Local share of organ procurement	579 (47.9)	56 (44.8)	0.57
Distance procurement from LT center, km	55 (10–226)	59 (10.226)	0.46
Use of plane for procurement	254 (21.0)	21 (16.8)	0.30

Abbreviations: GGT, gamma-glutamyl transferase; *n*, number; Q1–Q3, first–third quartile; HCC, hepatocellular carcinoma; HCV, hepatitis C virus; HBV, hepatitis B virus; NASH, non-alcoholic steatohepatitis; MELD, model for end-stage liver disease; CIT, cold ischemia time.

**Table 3 jcm-12-04744-t003:** Multivariable logistic regression analyses for the identification of donor- and procurement-specific risk factors for 90-day post-transplant graft loss: donor GGT introduced in the model as continuous or dichotomous variable (cut-off 160 IU/L).

Variable	Beta	SE	Wald	OR	95% CI		*p*
					Lower	Upper	
Donor GGT as continuous variable *							
MELD	0.04	0.01	24.11	1.04	1.03	1.06	<0.0001
Donor total bilirubin peak (mg/dL)	0.28	0.07	17.74	1.33	1.16	1.51	<0.0001
Split/partial liver	1.18	0.40	8.85	3.24	1.49	7.04	0.003
Donor history of alcoholic abuse	0.74	0.28	6.99	2.10	1.21	3.64	0.008
Donor age, years	0.01	0.01	6.55	1.01	1.003	1.02	0.01
VAS	0.003	0.001	5.01	1.003	1.00	1.01	0.03
Donor logGGT value (IU/L)	0.38	0.18	4.54	1.46	1.03	2.07	0.03
Donor GGT as dichotomous variable (160 IU/L) **							
MELD	0.04	0.01	23.11	1.04	1.03	1.06	<0.0001
Donor GGT value ≥ 160 IU/L	0.64	0.24	7.49	1.90	1.20	3.02	0.006
Split/partial liver	1.06	0.39	7.28	2.89	1.34	6.25	0.007
Donor history of alcoholic abuse	0.73	0.28	6.90	2.07	1.20	3.56	0.009
Donor age, years	0.01	0.01	5.26	1.01	1.00	1.02	0.02
VAS	0.003	0.001	4.67	1.00	1.00	1.01	0.03
Donor total bilirubin peak (mg/dL)	0.12	0.08	2.06	1.12	0.96	1.31	0.15

* −2 log-likelihood = 1045.77; Hosmer–Lemeshow, *p* = 0.77; ** −2 log-likelihood = 1063.51; Hosmer–Lemeshow *p* = 0.78; Variables initially included in the models: donor age; MELD; procurement LT center distance in Km/100; use of airplane for the procurement; split/partial liver; anoxia as cause of donor death; cerebrovascular accident as cause of donor death; donor history of diabetes; donor history of tobacco abuse; donor history of alcoholic abuse; donor anticore positivity; donor cardiac arrest before procurement; VAS; donor AST peak (IU/L); donor ALT peak (IU/L); donor total bilirubin peak (mg/dL); donor logGGT peak (IU/L) (first model); or donor GGT ≥ 160 IU/L (second model). Abbreviations: SE, standard error; OR, odds ratio; CI, confidence intervals; D-MELD, donor age model for end-stage liver disease; GGT, gamma-glutamyl transferase; VAS, vasoactive score; AST, aspartate transaminase; ALT, alanine transaminase.

**Table 4 jcm-12-04744-t004:** Improvement in model accuracy adding donor GGT to current criteria for evaluation of 90-day graft loss after liver transplantation.

Variables	K	−2LL	AIC	Delta AIC
MELD + donor GGT	4	1087.75	2179.50	0.00
MELD	3	1094.06	2191.12	11.62
BAR + donor GGT	4	1089.84	2183.68	4.18
BAR	3	1096.03	2195.06	15.56
D-MELD + donor GGT	4	1092.03	2188.06	8.56
D-MELD	3	1098.29	2199.58	20.08
DRI + donor GGT	4	1107.42	2218.84	39.34
DRI	3	1113.61	2230.22	50.72

Abbreviations: K, number of parameters in the model; AIC, Akaike information criterion; −2LL, −2 log-likelihood; MELD, model for end-stage liver disease; GGT, gamma-glutamyl transferase; D-MELD, donor age model for end-stage liver disease; DRI, donor risk index; BAR, balance of risk.

**Table 5 jcm-12-04744-t005:** Multivariable logistic regression analyses for the identification of donor-specific risk factors for high donor GGT value (≥160 IU/L) and for donor macrovesicular steatosis > 30% (sub-group of biopsied donors: *n* = 461).

Variable	Beta	SE	Wald	OR	95% CI		*p*
					Lower	Upper	
Donor GGT value ≥ 160 IU/L *							
Donor ICU length of stay, days	0.07	0.01	26.40	1.07	1.05	1.10	<0.0001
Donor history of alcohol abuse	0.65	0.32	4.08	1.92	1.02	3.61	0.04
Donor age, years	−0.01	0.01	3.76	0.99	0.98	1.00	0.053
Donor macrovesicular steatosis >30% **							
Donor BMI	0.14	0.05	7.50	1.15	1.04	1.28	0.006
Donor ICU length of stay, days	−0.23	0.08	7.43	0.79	0.67	0.94	0.006
Donor GGT peak (IU/L)	0.004	0.002	4.04	1.004	1.00	1.01	0.044
Donor history of type 2 diabetes	0.97	0.53	3.36	2.63	0.94	7.39	0.07
Donor total bilirubin peak (mg/dL)	0.23	0.12	3.29	1.25	0.98	1.60	0.07
Donor history of arterial hypertension	−0.65	0.40	2.58	0.52	0.24	1.15	0.11

* −2 log-likelihood = 792.15; Hosmer–Lemeshow, *p* = 0.17; Variables initially included in the models: donor age; donor male sex; anoxia as cause of donor death; length of donor ICU stay; donor history of diabetes; donor history of arterial hypertension; donor history of tobacco abuse; donor history of alcoholic abuse; donor anticore positivity; donor hypotension before procurement; donor cardiac arrest before procurement; DCD; VAS. ** −2 log-likelihood = 227.44; Hosmer–Lemeshow, *p* = 0.39; Variables initially included in the models: donor age; donor male sex; African origin; anoxia as cause of donor death; cerebrovascular accident as cause of donor death; donor BMI; length of donor ICU stay; donor history of diabetes; donor history of arterial hypertension; donor history of tobacco abuse; donor history of alcoholic abuse; donor anticore positivity; donor hypotension before procurement; donor cardiac arrest before procurement; DCD; VAS; donor AST peak (IU/L); donor ALT peak (IU/L); donor total bilirubin peak (mg/dL); donor GGT peak (IU/L). Abbreviations: SE, standard error; OR, odds ratio; CI, confidence interval; GGT, gamma-glutamyl transferase; ICU, intensive care unit; BMI, body mass index; DCD, deceased cardiac donor; VAS, vasoactive score; AST, aspartate transaminase; ALT, alanine transaminase.

**Table 6 jcm-12-04744-t006:** Literature review on the prognostic impact of donor GGT values after transplantation.

Study	Ref	Year	Country	*N*	GGT IU/L Cut-Off	Endpoint	HR
Zhang et al.	[14]	2021	US	53,966 47,249	>200 <20	Graft discard 1-year graft failure	2.74 0.91
Braat et al.	[15]	2012	Eurotransplant	5723	NA *	Graft failure	1.06
Hoyer et al.	[16]	2015	Germany	678	NA *	EAD	1.002
Capelli et al.	[17]	2021	France	1152	170	90-day graft failure	NA
Present study	-	2023	Italy-Belgium	1335	160	90-day graft failure	1.90 **

* Donor GGT evaluated as a continuous variable; ** Odds ratio. Abbreviations: Ref, reference; N, number; GGT, gamma-glutamyl transferase; HR, hazard ratio; NA, not available; EAD, early allograft dysfunction.

## Data Availability

Tha data of the present study are not available due to privacy restrictions, but can be obtained after formal request to the corresponding author after approval of all the coauthors.

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
