# Peer review of "The Role of Donor Gamma-Glutamyl Transferase as a Risk Factor for Early Graft Function after Liver Transplantation"

_jcm, 2023, doi:10.3390/jcm12144744_

Round 1
Reviewer 1 Report
The manuscript entitled THE ROLE OF DONOR GAMMA-GT AS A RISK FACTOR FOR EARLY GRAFT FUNCTION AFTER LIVER TRANS-PLANTATION by Quirino Lai et al., aims to investigate whether high donor GGT values were correlated with reduced post-LT graft survival.
This is an interesting multicenter study including a large sample size. The study is very well designed, it is well written and has an excellent statistical analysis. The conclusions are supported by the results which are novel and add to the existing literature on this topic.
There are minor concerns that need to be addressed.
Table 1 is confusing, please separate recipient characteristics and donor characteristics in different tables.
Minor typos that need correction
minor typos that need to be corrected
Author Response
We thank Reviewer#1 for the positive comments. As suggested, we divided Table 1 into two tables reporting the data on donation (new Table 1) and recipients and transplantation (new Table 2).Reviewer 2 Report
In the manuscript, Lai and colleagues designed a retrospective international study investigating 1,335 adult patients receiving a first LT and detected the role of donor GGT as a risk factor for early graft function after LT. It is a valuable study and the results provided some positive evidence about the clinical predictive value of GGT in LT. Some suggestions about the study are mentioned below.
Major
1. HCC is a very risk factor for graft and recipient survival, so I think patient with HCC need to be excluded into the analysis.
2. I note that the last follow-up date was March 31, 2021. So how is the predictive value of GGT in a longer time after LT?
3. I noted that the index of donor GGT was detected at the time of procurement, please describe the detected time for the other indexes such us donor AST peak (IU/L), donor ALT peak (IU/L), donor total bilirubin peak (mg/dL), donor INR, and donor platelets.
Minor
1. Please add the IRB number of the study in the methods.
2. In the results, the first sub-title about the characteristics of the cohort is missing.
Minor editing of English language required.
Author Response
Major
- HCC is a very risk factor for graft and recipient survival, so I think patient with HCC need to be excluded into the analysis.
Authors' response: We thank Reviewer#2 for the comment. HCC is a risk factor for sure a risk factor for median- and long-term patient survival. However, the impact of HCC in early (3-month) survival is neglectable. None of the reported cases has a recurrence or a tumor-related death within at least six months after transplantation. Therefore, we decided not to exclude the HCC cases from the present analysis.
- I note that the last follow-up date was March 31, 2021. So how is the predictive value of GGT in a longer time after LT?
Authors' response: We thank the Reviewer for the comment. Our intent was not to investigate the impact of donor GGT on median- and long-term survivals. As reported in Table 6, typically, donor GGT is explored in terms of very early or 90-day impact. Consequently, although our patients have reached an extended follow-up, we decided not to explore the long-term effects of GGT and the database was created and collected with this intent. If the Reviewer thinks this information is relevant to the study, we can contact the centers requiring the last available follow-up for each patient, intending to explore the 1-year and overall patient- and graft-related impact of donor GTT also on the long-term.
- I noted that the index of donor GGT was detected at the time of procurement, please describe the detected time for the other indexes such us donor AST peak (IU/L), donor ALT peak (IU/L), donor total bilirubin peak (mg/dL), donor INR, and donor platelets.
Authors' response: We thank Reviewer#2 for the relevant comment. We better clarify how we collected the donors' blood tests. In detail, all the donor blood tests collected during the ICU stay were evaluated, and the peak values were collected. In the case of GGT, the last available value was considered. The donor GGT peak value was not collected due to many missing data. We added this sentence in the Material and Methods part.
Minor
- Please add the IRB number of the study in the methods.
Authors' response: We thank Reviewer#2 for the suggestion. We added the approval number as required.
- In the results, the first sub-title about the characteristics of the cohort is missing.
Authors' response: We thank Reviewer#2 for the comment. We added the sub-title as required.
Reviewer 3 Report
The manuscript titled "THE ROLE OF DONOR GAMMA-GT AS A RISK FACTOR FOR EARLY GRAFT FUNCTION AFTER LIVER TRANSPLANTATION" investigates the potential risk associated with donor gamma-glutamyl transferase (GGT) levels at the time of organ procurement in liver transplantation (LT). The study, conducted in a large European LT cohort, aims to establish a relationship between donor GGT and the loss of graft within 90 days post-transplantation. The results indicate that donor GGT is an independent predictor of graft loss, with higher rates observed in cases where GGT levels are ≥ 160 IU/L. Furthermore, the inclusion of donor GGT in existing predictive scores improves their accuracy. However, further research is necessary to understand the mechanisms that contribute to the association between GGT and poor LT outcomes.
Overall, the manuscript effectively presents the study's objectives, methods, and results in a clear and concise manner. However, there are a few aspects that could benefit from additional clarification and discussion.
Firstly, while the study establishes a correlation between donor GGT and graft loss, the specific mechanisms underlying this relationship remain unclear. It would be beneficial to explore possible explanations or hypotheses that could shed light on this association. The authors appropriately acknowledge the need for prospective studies to further investigate this topic.
Secondly, it is important to discuss the potential clinical implications of these findings. Understanding the impact of donor GGT levels on graft function can have significant implications for pre-transplant evaluation and risk stratification. Exploring the practical applications of this discovery would enhance the manuscript's relevance.
Lastly, based on the current results, further exploration of the underlying mechanisms and clinical implications would strengthen the overall impact of the study. Continuation of research in these areas would contribute to a better understanding of the role of donor GGT in liver transplantation and potentially lead to improved outcomes for transplant recipients.
Author Response
1) Firstly, while the study establishes a correlation between donor GGT and graft loss, the specific mechanisms underlying this relationship remain unclear. It would be beneficial to explore possible explanations or hypotheses that could shed light on this association. The authors appropriately acknowledge the need for prospective studies to further investigate this topic.
Authors' response: We thank Reviewer#3 for the comment. In the Discussion, we tried to report the poor literature on the mechanisms justifying the correlation between poor graft function and high donor GGT. In detail, GGT could correlate with macrosteatosis, longer cold ischemia time, and longer donor ICU stay. We reported the following sentence: "We can only assume that different mechanisms might be involved, like an increased risk of infection, hemodynamic instability, and vasoconstriction derived from inotropic drugs, all concurring in a deleterious ischemic effect. Further studies are needed to clarify these aspects better."
2) Secondly, it is important to discuss the potential clinical implications of these findings. Understanding the impact of donor GGT levels on graft function can have significant implications for pre-transplant evaluation and risk stratification. Exploring the practical applications of this discovery would enhance the manuscript's relevance.
Authors' response: We thank Reviewer#3 for this relevant consideration. We totally agree with the comment. We added the following sentence in the Discussion part: "The potential clinical implications of the findings reported in the present study should be of relevance in terms of pre-transplant evaluation and risk stratification. In fact, donor GGT is typically not used as a selection parameter for donor graft quality, and many scores used in the selection process avoid to incorporate it. On the opposite, our study showed that the integration of donor GGT into commonly used scores plays a role in improving the risk stratification of the recipients. Therefore, a proposal for further exploring donor GGT in this setting should be of great relevance."
3) Lastly, based on the current results, further exploration of the underlying mechanisms and clinical implications would strengthen the overall impact of the study. Continuation of research in these areas would contribute to a better understanding of the role of donor GGT in liver transplantation and potentially lead to improved outcomes for transplant recipients.
Authors' response: We thank Reviewer#3 for this relevant consideration. We considered the Reviewer's comment as a perfect conclusion of our article, so we decided to incorporate it into our text, because it exactly resumes our idea on the relevance of this topic in the field of transplantation.
Reviewer 4 Report
Abnormal liver values, high grade of macrosteatosis and the history of significant alcohol use are the main reasons for discarding liver-graft offers in many European countries.
The decision to accept or decline a liver graft for transplantation is typically made by a multidisciplinary transplant team based on their experience and center-based protocols. These protocols may take into account various liver function parameters, including GGT, as well as other factors such as organ quality, donor characteristics, and recipient needs. The specific cutoffs or acceptable ranges for GGT levels can vary among different transplant centers and depend on their individual protocols and expertise. The acceptability of a liver for transplantation is determined through careful assessment of multiple parameters. These parameters include liver function tests, imaging studies, donor history, and biopsy results. The specific cutoff for GGT, if any, may also depend on the recipient's condition, and overall organ availability. The GGT cut-off value for detecting of a harmful grade of macrovesicular steatosis of 60% has been calculated as 142 U/L with a good accuracy (Johanna Savikko et al. Gamma-glutamyltransferase predicts macrovesicular liver graft steatosis - an analysis of discarded liver allografts in Finland. Scand J Gastroenterol 2023. 58 :412-416).
The cutoff value of 160 IU/L reported in this study seems very high. In this retrospective study design, inherent biases and limitations are declared by the authors. A critical note is the lack of clinical significance and practical implications since the marker of a GGT level higher than 160 IU/L could be of clinical value only when incorporated and proposed into existing predictive models.
My questions are:
1. There are 4 European centers in this study. All 4 centers have a common protocol in accepting donors who have such high (> 160 IU/L) GGT values? And under what conditions are these values considered acceptable by the four centers? High GGT levels and alcohol abuse are considered indications for a liver graft biopsy evaluation in most centers. Were all these donors carefully evaluated for histologic evaluation before a liver transplant to evaluate the percentage of liver macrosteatosis and exclude NAFLD or NASH? In table 1, 13 patients with alcohol abuse and GGT > 160 IU/L are reported. Is there anything you could add to the discussion about these high-risk donors?
2. It appears that 9.4% of donors (122 donors) had GGT levels > 160 IU/L, which is an important percentage. Can you explain under which conditions these 122 donors were considered an “acceptable risk” for liver donation and why 13 of them with alcohol abuse were also utilized?
3. Possible bias for generalizability: This study included 4 large European cohorts (3 in Italy and 1 in Belgium). The finding of an increased rate of donors with a high GGT value may not be similar in other European regions. Donor GGT can be a very significant risk factor for graft dysfunction after liver transplantation. This depends on the presence of other associated risk factors, namely varying degree and rate of of alcohol use, different presence of NAFLD in the population, and different rate of high BMI. All these factors might vary in various demographic and clinical contexts. Could you please split how many of these GGT values have been used in the 4 different centres?
4. Are living donors’ liver transplantations included in the study? Is the rate of LDLT similar in all four transplant centers?
5. Independent Predictor of Graft Loss: This study establishes that donor gamma-glutamyl transferase (GGT) has an independent role as a predictor of graft loss, both when analysed as a continuous variable and when dichotomized using a cut-off value. It seems that this data does not add anything very novel to what has been generally reported by many other transplant centres worldwide. As you mention a recent French study has outlined the role of GGT and alcohol abuse as devastating risk factors for poor outcomes after liver transplantation. This should be clearly outlined in the discussion.
Author Response
My questions are:
- There are 4 European centers in this study. All 4 centers have a common protocolinaccepting donors who have such high (> 160 IU/L) GGT values? And under what conditions are these values considered acceptable by the four centers? High GGT levels and alcohol abuse are considered indications for a liver graft biopsy evaluation in most centers. Were all these donors carefully evaluated for histologic evaluation before a liver transplant to evaluate the percentage of liver macrosteatosis and exclude NAFLD or NASH? In table 1, 13 patients with alcohol abuse and GGT > 160 IU/L are reported. Is there anything you could add to the discussion about these high-risk donors?
Authors' response: We thank Reviewer#4 for this relevant question. First, we added a sentence clarifying that no specific protocols exist in the involved centers concerning donors with high GGT values. Typically, the selection of the donors is performed case by case: in the presence of multiple worrisome aspects, the biopsy should be considered. However, a specific protocol does not exist, and the surgeon performs a biopsy "a la demande" during the procurement process. We added a sentence in Discussion, reporting the number of high-GGT cases receiving a biopsy. Unfortunately, the sample size of our study limited the opportunity to investigate more in detail the role of biopsy in specific subclasses like the donors with a previous history of alcoholic abuse. We only suggested that the use of biopsy should be a safe approach in this case.
- It appears that 9.4% of donors (122 donors) had GGT levels > 160 IU/L, which is an important percentage. Can you explain under which conditions these 122 donors were considered an "acceptable risk" for liver donation and why 13 of them with alcohol abuse were also utilized?
Authors' response: We thank Reviewer#4 for this question. Donor GGT does not represent a parameter typically adopted for excluding a graft from the donation process. On the contrary, many studies do not consider donor GGT a relevant issue to investigate in this setting. As for our approach, we retrospectively investigated a large population of cases in which a relevant percentage of cases (approximately 10%) showed a high GGT value. In all the cases, the surgeon decided if a biopsy was necessary after the graft macroscopic evaluation. In 54/125 cases (43.2%), a biopsy was performed before the clamping time, and in all the selected cases, the biopsy showed an acceptable graft quality. The biopsy was performed in all 13 cases of concomitant history of alcohol abuse. We added a sentence in the Discussion reporting these data.
- Possible bias for generalizability: This study included 4 large European cohorts (3 in Italy and 1 in Belgium). The finding of an increased rate of donors with a high GGT value may not be similar in other European regions. Donor GGT can be a very significant risk factor for graft dysfunction after liver transplantation. This depends on the presence of other associated risk factors, namely varying degree and rate of of alcohol use, different presence of NAFLD in the population, and different rate of high BMI. All these factors might vary in various demographic and clinical contexts. Could you please split how many of these GGT values have been used in the 4 different centres?
Authors' response: We agree with the Reviewer. We added a sentence in the limitation part, clarifying the possible bias. Nevertheless, no difference was reported in the four centers in terms of the percentage of high-GGT cases used in the four centers. In detail:
PTV Rome: 25.8%; Catholic Rome: 24.5%; Sapienza Rome: 27.0%; Gent: 22.7%.
- Are living donors' liver transplantations included in the study? Is the rate of LDLT similar in all four transplant centers?
Authors' response: We thank Reviewer#4 for this relevant aspect. We only considered deceased donors. We better clarified this aspect by adding the following sentence in Materials and Methods: "All the cases investigated included only LTs performed using grafts from deceased donors. No living donors or domino transplants were considered in the analysis."
- Independent Predictor of Graft Loss: This study establishes that donor gamma-glutamyl transferase (GGT) has an independent role as a predictor of graft loss, both when analysed as a continuous variable and when dichotomized using a cut-off value. It seems that this data does not add anything very novel to what has been generally reported by many other transplant centres worldwide. As you mention a recent French studyhas outlinedthe role of GGT and alcohol abuse as devastating risk factors for poor outcomes after liver transplantation. This should be clearly outlined in the discussion.
Authors' response: As the Reviewer can see in the Table dedicated to the Literature review, only two studies specifically explored the 90-day graft loss after LT, namely the French one and ours. All the other studies were based on different outcomes (i.e., graft discard, EAD, and overall graft failure). As reported by the Reviewer in a previous question, the risk of bias for generalizability exists. This limit is present in our study and the French one. Consequently, we think that our study adds something novel to the literature, exploring different geographical areas and using a different statistical approach to previous studies.
Round 2
Reviewer 4 Report
All reviewer's questions and comments have been satisfactorily answered by the authors.